# Deep Learning for Counting People from UWB Channel Impulse Response Signals

**DOI:** 10.3390/s23167093

**Published:** 2023-08-10

**Authors:** Gun Lee, Subin An, Byung-Jun Jang, Soochahn Lee

**Affiliations:** School of Electrical Engineering, Kookmin University, Seoul 02707, Republic of Korea; leegun4488@kookmin.ac.kr (G.L.); nesquiq@kookmin.ac.kr (S.A.); bjjang@kookmin.ac.kr (B.-J.J.)

**Keywords:** ultra-wideband, people counting, deep neural networks

## Abstract

The use of higher frequency bands compared to other wireless communication protocols enhances the capability of accurately determining locations from ultra-wideband (UWB) signals. It can also be used to estimate the number of people in a room based on the waveform of the channel impulse response (CIR) from UWB transceivers. In this paper, we apply deep neural networks to UWB CIR signals for the purpose of estimating the number of people in a room. We especially focus on empirically investigating the various network architectures for classification from single UWB CIR data, as well as from various ensemble configurations. We present our processes for acquiring and preprocessing CIR data, our designs of the different network architectures and ensembles that were applied, and the comparative experimental evaluations. We demonstrate that deep neural networks can accurately classify the number of people within a Line of Sight (LoS), thereby achieving an 99% performance and efficiency with respect to both memory size and FLOPs (Floating Point Operations Per Second).

## 1. Introduction

Ultra-wideband (UWB) technology is a wireless communication protocol that enables short-distance communication through high-frequency radio waves. While it first emerged in the 1970s in the United States for military applications, commercialization began in the 2000s after military restrictions were lifted [1]. Despite its advantages, UWB technology has not garnered as much attention as other wireless communication technologies like WiFi and Bluetooth, primarily due to its lower competitiveness in areas such as production cost [2,3]. But the adoption of the High-Rate Pulse Repetition Frequency (HRP) and the establishment of the IEEE 802.15.4-2015 [4] standard has helped the potential of UWB technology being recognized.

UWB technology has many advantages, including high data transfer speeds, low power consumption, and a resistance to interference from other wireless signals [5]. The utilization of higher frequency bands, in contrast to other wireless communication protocols, enhances the capability to accurately determine locations. UWB technology features a bandwidth of 500 MHz, and its narrow pulse width of 2 ns can be leveraged to obtain precise distance measurements. By correlating UWB symbols, multiple channel impulse responses (CIRs) can be obtained, and by statistically analyzing the slight variations in these CIRs, a distance resolution of several centimeters or less can be achieved [6]. This feature is useful for indoor positioning systems, asset tracking, and other location-based services.

These characteristics of UWB technology can also be applied for estimating indoor occupancy, that is, to determine the number of people in a given space [7,8,9,10]. This application is related to building management, optimizing resource allocation, as well as in enhancing safety and security in public places such as airports, shopping malls, and stadiums. For example, the use of people-counting technologies can greatly benefit environments such as retail stores or public transportation systems. In the context of a store, it can provide real-time data on crowd density, helping to maintain a safe environment by avoiding overcrowding. In this work, we provide experimentation on a small number of people, which may be applied to cases such as measuring elevator capacity.

Imaging technology has been traditionally used to determine the number of people indoors, but it can be challenging to accurately count the number of people in poorly lit areas or in places where people are partially obstructed by objects or other people. Moreover, these technologies carry inherent risks, including potential infringements on personal privacy and possibilities for unintentional disclosure of personal information.

Using radar technology requires expensive high-output and high-performance radar systems, which are not price-competitive [11]. And while Bluetooth or WiFi can be used to determine the presence of people, they may struggle to accurately recognize the number of individuals present [12]. Compared to Bluetooth or WiFi, UWB is less susceptible to the data attenuation caused by the presence of multiple people or objects due to its wider channel bandwidth and lower sensitivity to object interference.

In recent works [8,9], methods have been developed to estimate the number of people in a room based on the waveform of the CIR from UWB transceivers. In these methods, statistical methods such as simple thresholding [8] or analysis of the singular values of the matrix of temporal CIR differences [9] are used for estimation. While more recent machine learning methods, such as deep neural networks, have been applied to impulse radio (IR)-UWB radar signals [13,14,15,16], there has been no similar method for UWB transceiver CIR signals. Compared to radar, UWB transceivers have advantages in that they may be used together within a communication system, and that they may have less power consumption.

Thus, in this paper, we apply deep neural networks to UWB CIR signals for estimating the number of people in a room. We especially focus on empirically investigating various network architectures for classification from single UWB CIR data, as well as on various ensemble configurations. We present our processes for acquiring and preprocessing CIR data, our designs of the different network architectures and ensembles that were applied, and the comparative experimental evaluations. An ensemble of two models and four CIR signals as the best trade-off between performance and efficiency, achieved a 99% accuracy in 321,000 k FLOPs with 6566 k parameters.

## 2. Previous Work

Various works have been proposed in terms of applying UWB technology for detecting humans. As the task is involved with sensing, many works have applied IR-UWB radar as the signal source. In the work by Yang et al. [14], a method for dense people counting by using the hybrid features from IR-UWB radar was proposed. In the work by Lee et al. [17], a method for motion recognition through IR-UWB radar was explored. In the work by Kalyanaraman et al. [18], 14 UWB nodes were installed to determine whether a car door was open and if a person was present. A method of counting people in a wide area using IR-UWB radar was introduced in Choi et al. [11]. In Pham et al. [13], the use of convolutional neural networks (CNN) for people counting on IR-UWB was proposed. Human activity recognition using UWB CIR and Wi-Fi CSI was compared in Bocus et al. [19]. In Moro et al. [15], a variety of machine learning methods, including decision trees, as well as feed-forward, recurrent, and autoencoder neural networks, were evaluated in how they dealt with the problem of non-line-of-sight human detection. And in the work by Choi et al. [16], a customized deep neural network structure combining CNN and LSTM, a type of recurrent neural network, for people counting from UWB radar signals was proposed.

Many other works have been proposed that apply the CIR from UWB transceivers as the input signal. In Mohammadmoradi et al. [8], simple thresholding on the CIR signal was used to estimate the number of people in a room. In the method by De Sanctis et al. [9], the features computed from the singular value decomposition (SVD) of the time variation of the CIR were used as the input for Naive Bayes classifiers and decision trees for people counting. Jang et al. [10] further extended this work on the additional datasets achieved by hardware meeting the latest IEEE 802.15.4-2015 [4] standard. More recently, Sung et al. [20], proposed a method for accurately locating individuals indoors through using deep learning on UWB CIR input data. The work in this paper also deals with UWB CIR data for people counting, but it is focused on determining the best neural network structures and ensemble configurations for maximizing the estimation accuracy.

## 3. UWB Dataset

### 3.1. UWB Data Acquisition

To collect data, we used the DWM3000 module from Qorvo (Greensboro, NC, USA), which is the latest HRP UWB module supporting the IEEE 802.15.4z-2020 standard [21]. This module allows for selective frequency channel settings that are based on domestic communication standards. For our experiment, we set the module to channel 9. To connect the module to the PC, we used Nordic’s NRF-52840 microcontroller (Nordic, Trondheim, Norway).

Since the CIR is greatly influenced by the surrounding environment, efforts were made to create the same environment as much as possible, other than the variable for the number of people. The experimental environment was set up as depicted in Figure 1 when acquiring the CIR generated during transmission and reception between the two modules.

Room dimensions were 12 m × 8 m × 2.6 m. In a Line of Sight (LoS) environment, the two modules were placed 4 m apart in a straight line, each positioned 1 m above the floor and fixed using a tripod.

When acquiring the data, continuous communication was established between the two UWB modules. The test subjects were adult males and females. The number of people in the environment was then varied from zero to five, and the corresponding CIR signals were measured. To diversify the data, we varied the height, body shape, gender, and other factors for each person, as well as the freeze variations to the basic environment. And data were acquired for when the people remained stationary and when they moved and acted freely. The CIR signal was collected at a rate of two instances per second. In each setting, 500 CIRs were measured in a LoS environment, and then the number of people was gradually increased up to five, resulting in 3000 CIRs. We measured data for 11 different settings, 4 with static subjects and 7 with moving subjects, resulting in a total of 33,000 CIRs.

In the following Figure 2 and Figure 3, we present visualizations of the CIR waveforms for static and moving peoples, respectively. We observed that, as the number of people grew, the variance of waveforms also increased, and this was likely due to the added reflected waves from the additional individuals. On the other hand, we observed that the motion of the subjects did not seem to affect the CIR significantly. We believe that this is due to the characteristic high frequency of the UWB signal.

Signals with a higher index tend to show more variation than those with a lower index. This is primarily due to the nature of signal wave propagation. Lower index signals are predominantly formed by direct waves, which have a straightforward path. In contrast, higher index signals are largely formed by indirect waves that reflect off walls or objects, causing a greater degree of variation.

### 3.2. UWB CIR Data Format

A UWB pulse width of 1 ns provided a distance resolution of 30 cm. It was possible to increase this resolution to several centimeters by transmitting UWB symbols repeatedly and by detecting changes in the CIR values through slight differences in the internal clocks of the hardware. The resulting CIR data contained 64 subtle differences, and a resolution of 1/64 ns was achieved through a leading edge detection (LED) algorithm that leverages the data statistics. Often a proprietary LED algorithm is applied by the manufacturer of the UWB IC, but the CIR obtained through this process can be accessed through the SDK. In over 1000 CIR points, the first correlation point that surpasses a specific threshold is called the FP_index. Based on that point, we extracted 64 subsequent values of the CIR. Since the UWB receiver has an IQ demodulator structure, real and imaginary components were acquired within the CIR signal as follows:(1)$$CIR^{r}=\left[\alpha_{1},\alpha_{2},\ldots ,\alpha_{64}\right],\;CIR^{i}=\left[\beta_{1},\beta_{2},\ldots ,\beta_{64}\right].$$

We applied the concatenated real and imaginary vectors as the input data for the neural network. To guide the network to learn the relative shape patterns of the CIR instead of the absolute values which may depend on the signal strength, we applied min–max normalization on $CIR^{r}$ and $CIR^{i}$, separately, as follows:(2)$${\bar{CIR}}_{k}=\frac{CIR_{k}-CIR_{Min}}{CIR_{Max}-CIR_{Min}}.$$

We defined the input data for all neural networks in this work as
(3)$$\bar{CIR}=\left[{\bar{CIR}}^{r},{\bar{CIR}}^{i}\right].$$

## 4. Neural Network Components for CIR Signals

### 4.1. Fully Connected Layer

The fully connected layer comprised a set of neurons that were connected parallel to the input, as shown in Figure 4a. This is the basic layer type for neural networks, and it constitutes the hidden layer for multi-layer perceptrons. Its operation was defined as
(4)$$X_{out}=\sigma \left(WX_{in}+b\right),$$
where $X_{in}$ and $X_{out}$ are the layer input and output, respectively; *W* is the weight parameters; *b* is the bias vector; and σ is the activation function. If the layer was applied to the input $\bar{CIR}$, *W* would form a 2D matrix of size N×128, where *N* is the number of neurons in the layer. This layer was also referred to as the densely connected layer, or the linear layer.

### 4.2. Convolutional Layer

The convolutional layer comprised a set of convolutional neurons, as shown in Figure 4b. The operation within this layer is defined as a convolutional function with kernel values as the weight parameters, as per the below:(5)$$X_{out}\left[p\right]=\sigma \left(\sum_{i=-m}^{m}X_{in}\left[p+i\right]\cdot w\left[i\right]\right).$$

Here, *p* represents a specific position in both the input and output signals. The symbol *w* is used to denote the convolutional kernel, while *m* is the size of the kernel.

In contrast to the conventional, fully connected neurons, shared weight parameters were applied to a local window of input data at every point to compute the output vector for each convolutional neuron. In the case of a 1D convolution layer, the computation proceeds in one direction using convolution operations, which were determined by the kernel size. Although the output dimension was adjusted by the kernel size parameter k, we included zero padding to ensure that the same dimension was consistently maintained.

### 4.3. Activation Function

Both fully connected and convolutional layers required an activation function to avoid the operations of multiple layers collapsing into a single operation. While various activation functions have been proposed [23], we applied the ReLU (rectified linear unit) [24] function for all layers except for the final layer of a network (as it is the most widely used). The ReLU function was defined as follows:(6)$$\begin{array}{c} y=max(0,x). \end{array}$$

For the final layer, we applied the softmax function introduced in [25], which transforms the output value into a probability that corresponds to each class by the following formula:(7)$$\begin{array}{c} y_{k}=\frac{exp(x_{k})}{\sum_{i=1}^{N}exp\left(x_{i}\right)}. \end{array}$$

### 4.4. Pooling Layer

Pooling layers pool multiple local input values together into a single output in a sliding window fashion that is similar to convolutional layers. Pooling layers are mostly used together with convolutional layers to compensate for the convolutional layers’ limitations that arise from local operations. As convolutional neurons may fail to generate output dimensions that encompass the global range of the input data, pooling was applied to increase the ’receptive field’. The two common types of pooling are max pooling and average pooling, which were introduced in [26,27,28] and are also presented in Table 1. Max pooling selects the maximum value from within the defined pool size, thus preserving the strongest feature response. On the other hand, average pooling calculates the average value within the pool, providing a smooth, low-resolution representation of the feature map.

### 4.5. Normalization Layer

Normalization layers operate by rescaling data based on the mean of a specific dimension. When significant variation occurs within this dimension, normalization layers can effectively enhance the model’s stability, learning speed, and overall performance.

In this work, we experimented with both batch normalization and layer normalization, as introduced in [29,30] and presented in Table 2. Batch normalization normalizes the output of a prior layer with respect to the size of the batch, while layer normalization normalizes across the features of the input instead of the batch size.

In these equations, β represents the shift parameter, ϵ is the small constant added for numerical stability, and γ denotes the scale factor.

### 4.6. Dropout Layer

In a deep network where data passes through numerous layers, there is a risk that the model can become overly complex and begin to overfit, essentially memorizing the training data. The dropout operation introduced in [31,32] addresses this by randomly ’dropping out’ (that is, setting to zero) a number of layer outputs during the training process.

Through this approach, each neuron becomes less dependent on the activations of others, gaining more independent significance. This results in a more generalized model, as it becomes less reliant on specific weights and inter-neuron connections. Essentially, dropout layers bolster the model’s resilience when confronted with new, unseen data, leading to improved generalization performance.

### 4.7. Loss Function

The loss function in a machine learning model quantifies the disparity between the predicted probability distribution and the actual data distribution. It evaluates the model’s performance, guides the optimization process, and varies according to the problem type. Furthermore, loss functions help guide model training, enhance performance evaluation, improve learning algorithms, and assist in preventing overfitting through regularization.

Among the various loss functions, Cross Entropy [33] Loss is commonly used in classification problems; thus, we chose it for our experiment. It quantifies the difference between the predicted probabilities and the true label distribution, thereby aligning the model’s predictions more closely with the actual labels to improve its classification accuracy.
(8)$$L=-\frac{1}{N}\sum_{i=1}^{N}\sum_{j=1}^{C}y_{ij}\log\left({\hat{y}}_{ij}\right)$$

In this equation, *L* stands for the loss we aim to minimize. *N* refers to the batch size of the dataset, while *C* signifies the total number of classes present in our classification problem. The term *ŷ* represents the predicted probability from the model, and *y* symbolizes the ground truth, indicating that a specific observation *i* belongs to a certain class *j*.

## 5. Neural Network Structures

### 5.1. Fully Connected Neural Network

A Fully Connected (FC) Neural Network, a specialized type of artificial neural network, is characterized by each neuron in a layer being connected to all neurons in the preceding and succeeding layers. This comprehensive interconnectivity equips the network with the capacity to decipher and learn from the complex and abstract patterns in the input data.

Figure 5a depicts the configuration of the Fully Connected Neural Network. In order to compare accuracy, experiments were conducted by varying the number of hidden layers from one to three, and by setting parameter values between 16 and 1600.

### 5.2. Convolutional Neural Network

A Convolutional Neural Network (CNN) is a specialized category of deep learning models, and it is primarily used for tasks such as visual image analysis and other data-rich domains. These networks are designed to automatically discern the spatial hierarchies of features from input data, making them particularly well-suited for tasks where the spatial arrangement of data is crucial.

Figure 5b illustrates the configuration of our Convolutional Neural Network. The performance of the network is influenced by the parameter values and kernel size within the convolutional layer, so we conducted experiments by varying these two parameters, as well as the number of layers. In our experiments, we adjusted the parameter values between 64 and 800, as well as varied the kernel size from 3 to 15.

### 5.3. Mix Neural Network

Our mixed neural network was designed by integrating a convolutional layer with a fully connected layer. As shown in Figure 5c, the configuration consists of a three-layer Convolutional Neural Network (CNN) followed by one or two fully connected layers. We conducted experiments where we varied the parameters of the fully connected layer and the kernel size in the CNN layers, while keeping the remaining parameters constant (or ’frozen’).

### 5.4. AlexNet Style Network

We constructed a network by adopting the AlexNet [34] model, which is designed for effective image classification. We transformed the existing 2D network layer into a 1D form, and conducted experiments by changing the parameters, kernel size, and structure. The configuration diagram of the AlexNet Style Network is shown in Figure 5d. The AlexNet Style Network is composed of five Conv Layers and two FC Layers. In the first Conv Layer, a wide range is brought in with a large kernel size, and after bringing a narrow range of features through the next small kernel size, the FC layer learns information. Between each layer, there is an operation to reduce or expand information through the max-pooling layer and the zero-padding layer.

### 5.5. ResNet Style Network

ResNet, or Residual Network [35], is a deep learning model that utilizes “skip connections” or “shortcuts” to bypass layers, effectively addressing the problem of vanishing gradients and allowing for the training of much deeper networks. In a similar manner, a ResNet-style model was implemented by converting existing 2D-type layers into 1D-type, and by incorporating skip connections. Experiments were conducted by varying the parameters, kernel size, and structure. The structure of the ResNet Style Network is shown in Figure 5e.

In the case of the ResNet Style Network, block unit convolution was performed, as depicted in Figure 6. A Res block consists of a Conv block and identity block. A Conv Block is used when the existing size and the size created through Conv are different, while an identity block is used when the sizes are the same.

## 6. Ensembling Model and Signal

### 6.1. Model Ensembling

Model ensembling operates by integrating multiple individual models, usually by methods such as voting or through averaging predictions. This approach is characterized by the ability to leverage the strengths of various models, thereby often enhancing overall performance. By employing model ensembling, we can increase the robustness and generalization capabilities of machine learning models, thus frequently leading to higher prediction accuracies.

We generated two to four instances of each model, with each instance being trained with slightly different parameter values. Subsequently, we combined their outputs and evaluated their collective performance on the test set.

### 6.2. Signal Ensembling

Signal ensembling operates by merging multiple signal data points, typically by using methods such as averaging or computing a weighted sum to produce a consolidated output signal. A defining characteristic of signal ensembling is its capacity to reduce noise and improve the quality of the signal by leveraging information from several similar signals. Employing signal ensembling aids in enhancing signal reliability, boosting the signal-to-noise ratio, and ultimately improving the accuracy of the tasks that are dependent on the signal.

We used our top-performing model to predict between 2 and 10 signals. Subsequently, we combined these output signals and evaluated their collective performance on the test set.

## 7. Experiments

### 7.1. Experimental Setting

We conducted our work in a Linux OS, which served as the fundamental programming environment, and used Python version 3.6.9. In configuring the network, tensorflow library version 2.3.0 was employed, with the ADAM optimizer utilized to train the network and used to minimize sparse categorical cross-entropy loss. Data pre-processing was performed using the sklearn library. To analyze the performance index, the entire dataset was divided equally into a train dataset (70%), a validation dataset (15%), and a test dataset (15%).

### 7.2. Baseline Method

We conducted experimental evaluations of three basic classification methods: the decision tree classifier, the Naive Bayes classifier, and the k-nearest neighbor (K-NN) classifier. We trained these classifiers on the raw CIR data vector, and the lower dimensional vector was constructed by applying Principal Component Analysis (PCA) reconstruction with only a portion of the principal components, as conducted in the method for face recognition [36].

With a single signal, the performance appeared to be similar between the PCA method and raw data, as shown in Table 3. However, the PCA method helped to reduce the dimensionality of the data, which decreased the computational cost of training.

As for the K-NN classifier, it achieved an accuracy of approximately 66%. Although this can be improved with multiple iterations and adjustments to the components, the initial level of accuracy was not strongly indicative of a good performance.

### 7.3. Network Structure Search

We conducted a series of experiments using various models to determine the optimal network structure for estimating the number of people based on the received signals. Initially, we trained the models using a structure with minimal layers, fixing the batch size to 32, and continued this until no further improvements were seen in the learning process.

Additionally, we employed the ReduceLROnPlateau method, which adjusts the learning rate by reducing it to 90% of the original rate if no improvements are detected over a certain number of epochs. If we encountered a point where the learning plateaued despite the learning rate adjustments, we implemented Early Stopping to terminate the training process.

We then restored the model that demonstrated the best performance during training. The results from the various network structures we tested, including accuracy, number of parameters, and the Floating Point Operations Per Second (FLOPs), are summarized in Table 4. This table presents the best performance achieved for each architecture.

Generally, we observed that the accuracy increased with the number of layers. However, when the parameter value was set to its maximum, the network tended to become overly complex. This complexity not only slowed down the computation process, but also ultimately lead to a decrease in accuracy. The optimal kernel size appeared to be 9 as this provided the best performance.

Overall, the Mixed Style Network exhibited a higher accuracy than both the Fully Connected Network and the Convolutional Network. However, this came with a significant increase in the number of parameters and FLOPs. The AlexNet and ResNet models also had a considerable number of parameters and FLOPs. Interestingly, despite being deeper than AlexNet, ResNet had fewer parameters.

### 7.4. Network Hyperparameter Tuning

To compare the performance of the various neural network structures, we needed to perform hyperparameter tuning for each model. This tuning was based on the model with the highest performance within each network structure.

Initially, we incorporated the norm layer, dropout layer, and pooling layer into the optimal network and compared the experimental results. We tested various configurations for the norm and pooling layers as previously mentioned. After identifying the network with the highest performance, we further optimized it by adjusting the dropout rate.

Subsequently, we conducted experiments using various batch sizes, specifically 16, 32, 64, 128, and 256. In the final step, we adjusted the learning rate to maximize the performance of the model. The results of these experiments are detailed in Table 5.

### 7.5. Implementation Ensembling of Model and Signal

#### 7.5.1. Model Ensembling

In our experiment, we ensembled both similar and different types of models to obtain our best result. We focused on two types of models with the aim of finding the optimal model ensemble.

We conducted experiments with Fully Connected (FC) Networks, which are known for their rapid processing speed due to low FLOPs. Additionally, we tested ResNet models, which—despite their deep architecture—are recognized for their low parameter count.

By ensembling these diverse models, each with varying complexities and architectures, we aimed to enhance the overall performance of the system.

Our experimental results, as shown in Table 6, demonstrated that the ensembling of models resulted in an approximate performance improvement of 2–3% compared to a single model. In the case of FC Networks, which are not as deep as other model architectures, we observed performance limitations even when multiple models were used. However, the inclusion of ResNet models in the ensemble led to a noticeable improvement in performance.

Table 7 shows the average estimation accuracy for different numbers of people. Interestingly, the accuracy for three and four people was lower than for five. This suggests that while the waveforms for three and four people are complex, they are more easily confused with each other, leading to decreased accuracy.

The ResNet model showed a superior performance in situations where the signal was highly variable, such as when there were four–five individuals. This indicates that deep neural networks can adeptly handle challenging conditions, thereby offering increased reliability amidst signal instability or noise.

We carried out experiments to investigate the variations in accuracy that were contingent on the number of individuals, and this was achieved by using both single and ensemble models. These experiments provided insights into how the selection of the model and the ensembling strategy could impact performance based on the number of individuals present.

The results underscore the benefits of model ensembling, particularly its capacity to augment performance beyond that of individual models in complex and noisy environments. While model ensembling enhances performance without extending processing time, as the models operate separately, it does increase the memory usage depending on the number of models utilized.

#### 7.5.2. Signal Ensembling

The traditional evaluation method, which took into account only a single CIR, was expanded upon to achieve a better performance by considering multiple CIRs. This approach significantly improved accuracy. However, it is worth noting that this improvement came with an associated increase in FLOPs, which was proportional to the number of CIRs used, thereby highlighting a key trade-off between performance and computational cost.

To obtain more detailed performance metrics, we calculated the error rate between the predicted and actual number of people using the Mean Square Error (MSE) method. We conducted similar experiments with the ResNet Style Network model, which demonstrated the highest performance in our tests.

Prior research [9,10] indicates that accumulating 100 CIRs and performing SVD can achieve excellent accuracy. However, this approach requires at least 30 s to accumulate the CIRs and carry out the SVD. While we followed the descriptions of prior research, using the three features of a second slope, an average slope, and the area under curve for training the classifiers, we observed that it underperformed with the neural-network-based classifiers.

Our experimental results, as detailed in Table 8, demonstrate that ensembling multiple signals simultaneously yields a substantial performance improvement. Specifically, when comparing the results of using two signals to using just a single signal, we observed a notable increase of approximately 8% in performance. As the number of combined signals increases, the system has the potential to reach a 100% accuracy score. However, a crucial factor to consider is the associated increase in processing time. While the memory usage remains unaffected, the processing time increases proportionally with the number of accumulated signals.

#### 7.5.3. Model Signal Ensembling

Our experiments focused on achieving the maximum possible accuracy in terms of estimating the number of people, while also concurrently enhancing the efficiency of model parameters and FLOPs. In this context, we found the strategy of integrating both model ensembles and signal ensembles highly effective. Nevertheless, an anticipated trade-off emerged: the enhanced accuracy was offset by an increased number of FLOPs, which was a direct consequence of the larger parameter set introduced by the ensembles.

To address this challenge, we concurrently processed both the model and signal ensembles, a strategy that resulted in high accuracy with a reduced count of parameters and FLOPs. We already observed that the FC Net model significantly reduced the number of parameters compared to other models. In contrast, the ResNet model demonstrated high accuracy in handling highly variable signals.

This observation prompted us to combine the advantages of both models, sequentially integrating them and combining between one and four signals. This innovative approach leveraged the strengths of the different model architectures, thus potentially resulting in more robust and accurate predictions.

In order to achieve the benchmark performance of 99%, a reliance solely on model ensemble techniques is not sufficient. We found it necessary to combine 10 signals in a signal ensemble to surpass this performance threshold. However, as shown in the Table 9, the best performance can be achieved by a combination of just four signals and two models. Additionally, the computational cost in terms of FLOPs for this configuration was less than half of that required when utilizing 10 signals. This demonstrates that with strategic selection and the combination of models and signals, high performance can be achieved more efficiently.

## 8. Limitation

In Figure 7, we illustrate the effect on different environments by the CIR waves, which is achieved by visualizing the mean and variance of the 100 CIR waveforms that were measured in various settings. With the environment depicted in Figure 1, we compare two versions, with and without a desk present in between the Rx and Tx in the top row of Figure 7 (denoted as environment-1 and environment-1+desk). It was visible that even the placement of the desk had a clear effect on the basic waveform. In addition to the impact of moving objects, the static elements within different environments, such as those observed in the two additional settings of the bottom row of Figure 1 (denoted environment-2 and environment-3), also considerably modified the CIR.

These variations indicate that training must be performed for each environmental condition. Furthermore, if there are moving objects within the environments, which further complicate the influence on the CIR, it will be desirable to perform training on the data achieved, which includes the motion of the objects. While this may incur initial costs, we believe that acquiring the ground truth number of people in the training data can be conducted by various methods, e.g., manual counting, motion sensors at entrances, etc.

We acknowledge that the experiments conducted in this paper are limited in the number of counted people. For applications such as maintaining room occupancy under a certain limit, it is likely that the number of people will be much higher. In future works, we plan to expand our system for the estimation of larger range of people, including experimentation on systems including multiple Rx and Txs.

## 9. Conclusions

In this paper, we presented the process of acquiring CIR data via UWB, and integrated it into an artificial intelligence model. The experiment involved training the AI model to recognize the changes in the CIR waveforms that corresponded to the number of people, and to estimate the headcount based on the experimental data.

The highest-performing model yielded an accuracy of 81.86% when using a single CIR to estimate the number of people. This accuracy improved significantly to 86.32% when we applied a model fusion technique, and impressively reached 100% with the use of consecutive CIRs. Furthermore, by combining multiple models and signals—specifically for just four CIR signals—we were able to achieve an accuracy of 99%. This approach proved to be more efficient than using only model ensemble or signal ensemble techniques.

This approach can be adapted based on the computational capacity and type of board used to implement the AI model. By integrating the AI model with UWB technology, we can expect advancements in smart home and security applications.

## Figures and Tables

**Figure 1 sensors-23-07093-f001:**
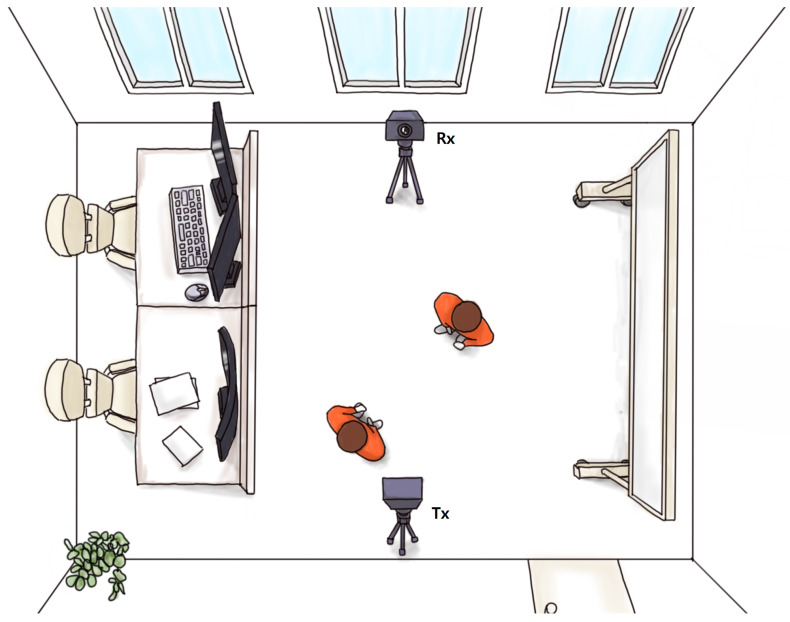
Laboratory Space Configuration. Experiment in the middle of room for decreasing signal reflect.

**Figure 2 sensors-23-07093-f002:**
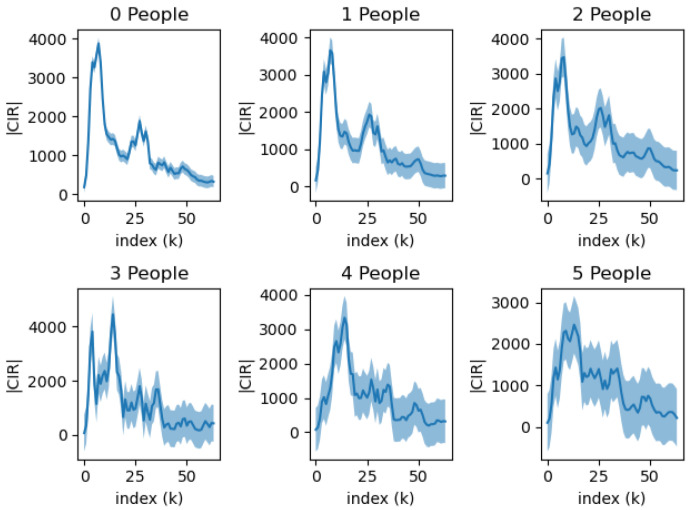
The change in CIR waveform depends on the number of stationary people between the Line of Sight (LoS). Each plot represents the mean and variation calculated from 500 individual CIR waveforms.

**Figure 3 sensors-23-07093-f003:**
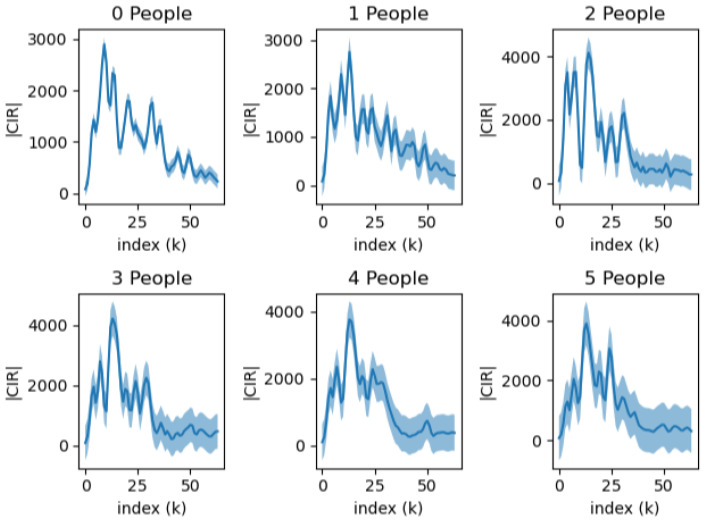
The change in CIR waveform depends on the number of individuals moving within the Line of Sight (LoS). Each plot represents the mean and variation calculated from 500 individual CIR waveforms.

**Figure 4 sensors-23-07093-f004:**
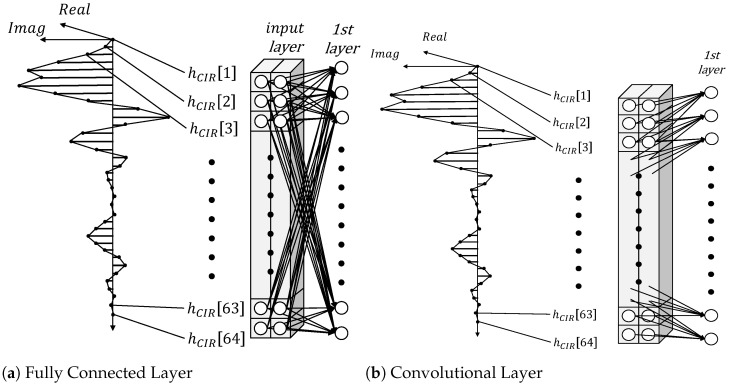
The format of CIR data and the configuration diagram for each layer [22]. $h_{CIR}\left[i\right]$ denotes the $i_{\mathrm{th}}$ element of the CIR input signal.

**Figure 5 sensors-23-07093-f005:**
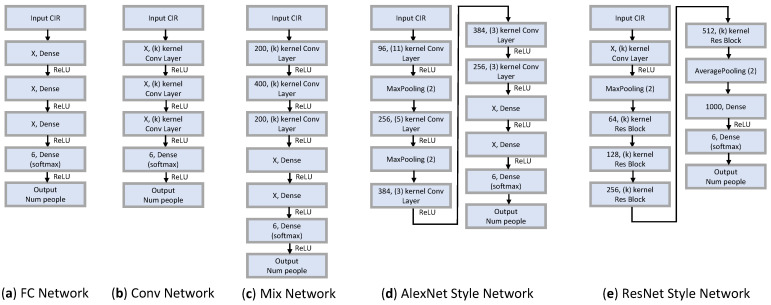
Network architectures explored in this work.

**Figure 6 sensors-23-07093-f006:**
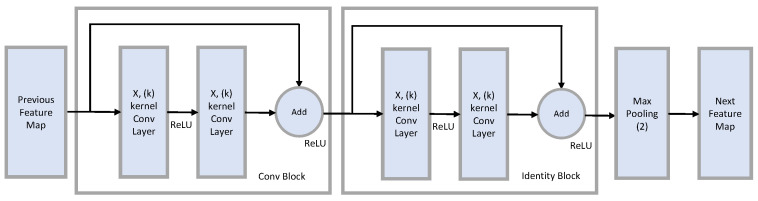
Diagram of Residual Block.

**Figure 7 sensors-23-07093-f007:**
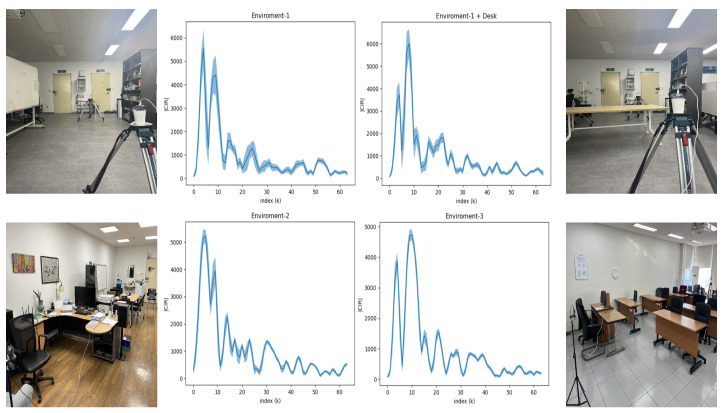
The change in CIR waveform depends on the environment. Each plot represents the mean and variation calculated from 100 individual CIR waveforms within the Line of Sight (LoS).

**Table 1 sensors-23-07093-t001:** Max Pooling and Average Pooling Equations.

Max Pooling Equation	Average Pooling Equation
$$\begin{array}{c} X_{out}\left[p\right]=max\left(X_{in}\left[p-m\right],\ldots ,X_{in}\left[p+m\right]\right) \end{array}$$	$$\begin{array}{c} X_{out}\left[p\right]=\frac{1}{2m+1}\sum_{i=-m}^{m}X_{in}\left[p+i\right]. \end{array}$$

**Table 2 sensors-23-07093-t002:** Batch Normalization and Layer Normalization Equations.

Batch Normalization Equations	Layer Normalization Equations
$$\begin{array}{c} \mu_{B}=\frac{1}{m}\sum_{i=1}^{m}x_{i}\;\;//\;\mathrm{mini-batch}\;\mathrm{mean}\\ \sigma_{B}^{2}=\frac{1}{m}\sum_{i=1}^{m}{\left(x_{i}-\mu_{B}\right)}^{2}\;\;//\;\mathrm{mini-batch}\;\mathrm{variance}\\ \hat{x_{i}}=\frac{x_{i}-\mu_{B}}{\sqrt{\sigma_{B}^{2}+\epsilon }}\;\;//\;\mathrm{normalize}\\ y_{i}=\gamma \hat{x_{i}}+\beta ={\mathrm{BN}}_{\gamma ,\beta }\left(x_{i}\right)\;\;//\;\mathrm{scale}\;\mathrm{and}\;\mathrm{shift} \end{array}$$	$$\begin{array}{c} \mu_{L}=\frac{1}{d}\sum_{i=1}^{d}x_{i}\;\;//\;\mathrm{layer}\;\mathrm{mean}\\ \sigma_{L}^{2}=\frac{1}{d}\sum_{i=1}^{d}{\left(x_{i}-\mu_{L}\right)}^{2}\;\;//\;\mathrm{layer}\;\mathrm{variance}\\ \hat{x_{i}}=\frac{x_{i}-\mu_{L}}{\sqrt{\sigma_{L}^{2}+\epsilon }}\;\;//\;\mathrm{normalize}\\ y_{i}=\gamma \hat{x_{i}}+\beta ={\mathrm{LN}}_{\gamma ,\beta }\left(x_{i}\right)\;\;//\;\mathrm{scale}\;\mathrm{and}\;\mathrm{shift} \end{array}$$

**Table 3 sensors-23-07093-t003:** Comparative analysis of the methods and classifiers with a single CIR signal.

Method	Classifier	Accuracy	Data Dim.
Raw data	Naive Bayes	37.68%	128
Raw data	Decision Tree	46.59%	128
Raw data	K-NN (K = 3)	66.16%	128
PCA	Naive Bayes	34.32%	57
PCA	Decision Tree	43.19%	24
PCA	K-NN (K = 3)	66.38%	57

**Table 4 sensors-23-07093-t004:** Summary of the network structures search results. (X and k correspond to each of the networks displayed in Figure 5).

Network Type	Layer Width X, Kernel Size (k), Res Block []	Accuracy	Parameters	FLOPs
FC Net	250	68.00%	33 k	67 k
FC Net	128-800	70.67%	124 k	248 k
FC Net	200-400-200	73.05%	187 k	374 k
Conv Net	400(13)	71.21%	164 k	1660 k
Conv Net	256(11)-400(11)	72.38%	1286 k	145,000 k
Conv Net	200(3)-400(9)-200(9)	76.12%	1518 k	185,000 k
Mix Net	200(3)-400(9)-200(9)-512	78.53%	7999 k	198,000 k
Mix Net	200(3)-400(13)-200(13)-256-128	79.68%	5392 k	273,000 k
Alex Net	96(11)-256(5)-384(3)-384(3)-256(3)-1024-2048	80.14%	9038 k	65,200 k
Res Net	4096(9)-[64(3)]-[128(3)]-[256(3)]-[512(3)]-1000	79.92%	5495 k	79,900 k

**Table 5 sensors-23-07093-t005:** Results of Network Hyperparameter Tuning.

Network	Hyperparameter					Acc.	Params.	FLOPs
	Norm	Dropout	Pool	Batch	LR			
FCNet	BN	0.2	X	64	1 $\times \;10^{-3}$	80.87%	190 k	381 k
ConvNet	X	0.3	X	128	1 $\times \;10^{-3}$	80.63%	1518 k	185,000 k
MixNet	X	X	X	64	5 $\times \;10^{-4}$	81.27%	5392 k	273,000 k
AlexNet	LN	X	AP	32	1 $\times \;10^{-4}$	81.31%	41,627 k	153,000 k
ResNet	BN	X	MP	32	1 $\times \;10^{-4}$	81.86%	6375 k	79,900 k

**Table 6 sensors-23-07093-t006:** Sample Results of 2-Network Ensembling.

Network	Acc.	Params.	FLOPs
FCNet + FCNet	81.57%	463 k	926 k
ResNet + ResNet	82.64%	12,751 k	160,000 k
ResNet + FCNet	83.27%	6566 k	80,300 k

**Table 7 sensors-23-07093-t007:** Average Estimation Accuracy for the Number of People.

Num of People	Single FC Network	Single ResNet	FC Ensemble	ResNet Ensemble
0	99.75%	99.51%	99.75%	99.63%
1	89.69%	92.72%	91.75%	92.60%
2	81.93%	82.18%	83.15%	84.72%
3	72.60%	72.48%	74.78%	74.78%
4	66.78%	68.72%	68.48%	72.00%
5	74.42%	75.51%	74.54%	77.93%
Mean	80.86%	81.85%	82.08%	83.61%
FLOPs	381 k	79,900 k	1140 k	240,000 k

**Table 8 sensors-23-07093-t008:** Results of Ensembling Multiple CIR Signals (Calculation of SVD FLOPS was taken from [37]).

Method	CIR Num	Accuracy	MSE	FLOPs
SVD & Naive Classifier	100	32.00%	1.8439	43,433 k
SVD & Decision Tree	100	60.00%	1.0392	43,433 k
SVD & K-NN (K = 3)	100	54.00%	1.1575	43,433 k
Neural Network	1	81.85%	0.6874	79,900 k
Neural Network	2	90.16%	0.3390	160,000 k
Neural Network	3	93.57%	0.1853	240,000 k
Neural Network	4	95.46%	0.1287	320,000 k
Neural Network	8	98.54%	0.0426	639,000 k
Neural Network	9	98.71%	0.0376	719,000 k
Neural Network	10	100 %	0.0000	799,000 k

**Table 9 sensors-23-07093-t009:** The distribution of accuracy varies as the number of people changes when using an ensemble of FC Net and ResNet.

Num of People	CIR Num 1	CIR Num 2	CIR Num 3	CIR Num 4
0	99.87%	100%	100%	100%
1	93.33%	99.27%	100%	100%
2	83.76%	93.44%	98.18%	99.02%
3	74.42%	88.83%	93.45%	95.63%
4	70.67%	84.71%	88.36%	95.14%
5	77.58%	91.02%	96.00%	98.05%
Mean	83.27%	92.88%	96.00%	98.98%
FLOPs	80,300 k	161,000 k	241,000 k	321,000 k

## Data Availability

The dataset used in this work is publicly available at: https://github.com/LeeGun4488/UWB_people_counting/tree/main/data%201d, accessed on 6 August 2023.

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
