# Peer review of "Deep Learning for Counting People from UWB Channel Impulse Response Signals"

_sensors, 2023, doi:10.3390/s23167093_

Round 1
Reviewer 1 Report
The authors present a method and experimental study for counting occupants of a room using UWB channel impulse response signals. Counting people in rooms is of particular interest for a number of applications, and having measurement modalities other than optical is of interest. As such, the paper makes a contribution to this field.
The paper requires a minor revision before acceptance. The work described in this paper, which is experimental in nature, seems to be a fairly limited case (small room and 0 to 5 occupants) and does not address all modalities that were mentioned of interest, for example, hidden and partially hidden people. In addition to these, what happens when objects in the room other than people move around?
The paper should provide better motivation for this small case (what is the concern: is the room occupied, is it close to the occupancy limit, etc.) and how it could be expanded to the more interesting cases of large rooms with many people (like conference rooms) and work spaces. The method is also not compared with other counting methods, like optical and IR, and what those accuracies are.
The conclusion is important that both model ensembles and signal ensembles is highly effective. But, it seems signal ensembles provides the bulk of the effectiveness. Please state what increase in accuracy is provided by each and then together. Perhaps the highly computationally expensive model ensembles could be relaxed. How will the computation expense scale with larger rooms and more people?
Some particular elements that could use clarification:
Figure 1: please provide dimensions for room size and antenna spacing.
Figure 2 and 3: it is not clear if these are single CIRs or averaged/collections
E.g. Table 3 and others: When the term accuracy is used, is this a simple binary value (count correct or not)? It could perhaps be interesting to state how far off the count was (say returned a value of 4 when 5 was expected, wish is less bad than returning a value of 1 when 5 was expected).
Overall, the paper is reasonably well written. However, there are several places with sentence fragments, i.e., last sentence of abstract (was there more needed?), lines 44-45, lines 275-276, etc.
Reviewer 2 Report
Some revisions need to be made, please look at the attached file. I would strongly suggest some additional explanation on why Deep Learning is necessary for counting people, since it appears that simple signal processing methods would a similar result with much less FLOPs.

Round 2
Reviewer 2 Report
All of the suggestions were addressed. I recommend publication.